# Exploring equity in audit and feedback trials: Secondary analysis of a systematic review

Zeenat Ladak[1,2‡]*, Camille Williams[2‡], Tolulope Ojo[1], Camille Renee[2],
Aranee Senthilmurugan[2], Thomas A. Willis[3], Victor C. Rentes[1], Armaghan Dabbagh[1],
Heather A. Shepherd[1], Tasneem Khan[3], Janyce Gnanvi[4], Mary Carter[5],
Ambreen Sayani[1,6], Lynne Moore[4,7], Aisha Lofters[1,2,6], Noah M. Ivers[1,2,6]

1 University of Toronto, Toronto, Ontario, Canada, 2 Women's College Hospital Institute for Health System Solutions and Virtual Care, Toronto, Ontario, Canada, 3 University of Leeds, Leeds, United Kingdom, 4 Université Laval, Québec, Canada, 5 University of Exeter, Exeter, United Kingdom, 6 Women's College Hospital, Toronto, Ontario, Canada, 7 Hôpital de l'Enfant-Jésus, Québec, Canada

‡ These authors are co-first authors on this work.
* zeenat.ladak@mail.utoronto.ca

## Abstract

### Background

One potential approach to eliminating or reducing health inequities for health systems is audit and feedback (A&F). A&F involves providing measurements of quality indicators to health professionals to support continuous quality improvement, and to increase clinicians' adherence to clinical practice guidelines. In theory, A&F could help direct efforts toward equity deserving sub-groups (e.g., gender-diverse individuals or those living with low-income) by highlighting factors that may place such sub-groups at higher risk of poor health outcomes. In cases where healthcare professionals can make adjustments to their practice or advocate for mitigating supports or services, A&F – when applied broadly – could help to address some health inequities. However, it is unknown whether and how A&F interventions are currently being used to support equity-oriented quality improvement. In this study, we sought to examine the extent to which trials evaluating A&F interventions address health equity.

### Methods

We conducted a secondary analysis of randomized controlled trials included in the latest Cochrane systematic review on the effects of A&F on professional practice, which included articles published up to 2020. We used the PROGRESS-Plus framework to consider the extent to which variables related to equity were examined in the trials. Based on extracted data, studies were categorized as not equity-oriented, equity-informed, or equity-focused.

**Data availability statement:** All relevant data are within the manuscript and its Supporting Information files.

**Funding:** This work was supported by an Ontario Graduate Scholarship, awarded to ZL.

**Competing interests:** The authors have declared that no competing interests exist.

## Results

Of the 271 articles included within this analysis, 44% of trials were classified as not equity-oriented (n = 120), 35% as equity-informed (n = 95), and 21% as equity-focused (n = 56). The proportion of equity-focused and informed trials increased over the timeline assessed. Only two articles described an equity-oriented framework approach. Only three articles explicitly reported how equity was embedded in their A&F process by highlighting factors including age, gender/sex, and substance use as part of the patient data presented in their feedback. The PROGRESS-Plus factors most commonly considered in the methods or analysis of the trials were age, insurance status, place of residence, and gender/sex.

## Conclusions

A&F trials rarely examine or report the extent to which equity issues inform trial design, A&F processes, analyses, and/or interpretations. Our findings suggest a need for future A&F trials that test explicit approaches to incorporating equity-related interventions to address health equity by helping healthcare professionals, teams, and organizations to be more aware of inappropriate discrepancies in care.

## Introduction

When health inequalities are due to avoidable factors (e.g., exposure to unhealthy living or working conditions), are systematically associated with social (dis)advantage, and are deemed unfair or unjust, they are considered health *inequities*. [1,2] To help understand the root causes of health inequities, scholars have developed frameworks outlining the relationships between health outcomes (e.g., life expectancy, mortality, disability) and the social determinants of health. [3] The social determinants of health include factors that may influence health outcomes based on the daily conditions in which people work and live. [4] The influence of the social determinants of health on health outcomes is complex and can take many forms such as reduced access to care for those living in rural regions, an increased risk of adverse health outcomes due to environmentally poor employment conditions, or stigmatizing patient-provider interactions for those who identify as gender or racially diverse individuals. [1,2] Frameworks suggest that social, economic, and political structures of society give rise to socioeconomic positions which stratify populations, resulting in systemic health inequities. [5] Consequently, it is not surprising that eliminating or reducing health inequities remains a daunting task for health systems and governments worldwide.

A focus on health equity as an essential component of high-quality care is a relatively recent development in health care; quality improvement initiatives, in general, have existed for decades. [6,7] One way to improve quality of care is through regional or national implementation of clinical practice guidelines. [8] These guidelines collate

the best evidence for clinical practice and outline factors that clinicians should consider at the point of care to better align their clinical decisions and behaviours with evidence-based practices. [8]

Audit and feedback (A&F) is often used to increase clinicians' adherence to clinical practice guidelines. [8] A&F involves providing measurements of quality indicators to health professionals, teams, and/or organizations to support continuous quality improvement [9]. A&F can improve quality of care when clinicians actively engage with the intervention and use it to support quality improvement efforts. [9] However, it is unknown whether and how A&F initiatives are currently being used to support equity-oriented quality improvement. If A&F initiatives are considering equity-oriented quality improvement, it is unknown which patient or health professional characteristics A&F developers consider to be relevant to equity or what frameworks are used to understand the relationship between health outcomes, sociodemographic factors, and socioeconomic factors. Further, there is evidence that implementation of clinical practice guidelines may contribute to health inequalities by disproportionately improving the health of those who are relatively health advantaged *more* than those who are relatively health disadvantaged (e.g., those living with low-income or gender-diverse individuals). [10,11] This makes it more difficult for those who need care the most to access quality care, as described by the inverse care law. [12] As such, it is imperative that A&F researchers examine whether and how their interventions are impacting health equity.

### Research question

To examine the extent to which A&F trials address health equity, we will answer the following questions:

1. What proportion of A&F trials are equity-oriented, that is, have an explicit focus on health inequities?

2. Which equity-oriented patient and/or health professional characteristics are being considered in these trials, and how?

3. What frameworks, theories, or conceptual approaches are being used in these trials to understand and address health inequities?

### Methods

We conducted this study in accordance with the Preferred Reporting Items for Systematic reviews and Meta-Analyses (PRISMA) reporting guidelines and specifically followed the PRISMA-Equity Checklist (S1 Table). [13]

### Study selection criteria

We conducted a secondary analysis of articles included in the latest update of the Cochrane systematic review on the effects of A&F on professional practice [9]. For that systematic review, studies were identified from the following databases: MEDLINE (Ovid), EMBASE (Ovid), CINAHL (EBSCO), the Cochrane Library (CENTRAL), clinicaltrials.gov, and WHO ICTRP (searched to February 2019 due to COVID-19 pandemic). The complete search strategy for that systematic review is available elsewhere and the study selection criteria are outlined in Table 1, with the exception of "intervention & domain" which reflects the domain of our secondary analysis, not the systematic review [9]. For this secondary analysis, we included all studies that were selected for the systematic review. In cases where an article reported on multiple primary outcomes, we considered the article for equity-orientation only once, as a whole, not for each outcome separately.

### Data extraction & analysis

A pre-determined data collection form was used to collect information on study design, participants, setting, interventions, outcomes, and results for the Cochrane review [9]. For this study, additional data extraction related to equity-oriented data was collected using a Microsoft Forms data collection form. The form was tested by four reviewers with three articles and modified as needed before extracting all included studies. Included studies were independently

**Table 1. Study Selection Criteria.**

| Criteria for Selection | Inclusion/Exclusion Criteria |
|---|---|
| **Participants/population** | Healthcare settings evaluating A&F interventions that target the behaviour and actions of healthcare professionals related to patient care. |
| **Intervention & domain** | Exploration or reduction of health inequities through A&F trials in healthcare settings |
| **Comparator(s)/control** | No intervention, usual care, or other quality improvement interventions (may include another type of A&F) to change clinicians' behaviour/actions |
| **Outcomes** | Objectively measured health professional/clinical practice outcomes, e.g., prescribing, test ordering, referrals. Studies that measure *only* patient health outcomes, knowledge/attitudes, or performance in a test situation were excluded. |
| **Study design** | Randomized controlled trials of any type (e.g., parallel, cluster, crossover) |

extracted by two reviewers in duplicate. Disagreements and discrepancies were resolved through discussion and/or by a third reviewer, as needed.

Data extraction included study characteristics, intervention descriptions, recruitment processes, findings, discussions related to each PROGRESS-Plus factor, and frameworks, theories, or conceptual approaches used to address health inequities. The PROGRESS-Plus framework is a comprehensive listing of such socially stratifying factors that may be related to inequities (S2 Table). [14] PROGRESS stands for place of residence, race/ethnicity/culture/language, occupation, gender/sex, religion, education, socioeconomic status, and social capital. [14] "Plus" factors include other personal characteristics that may attract discrimination (e.g., age, disability, substance use), features of relationships (e.g., smoking parents), time-dependent relationships (e.g., leaving hospital) and any additional context-specific factors that may indicate disadvantage. [15] The PROGRESS-Plus framework was developed by the Campbell and Cochrane Equity Methods Group as a tool to systematically identify and analyze health disparities in public health and policy research. [14–16]

## Data synthesis

The complete data extraction form is available in the supplemental information (S1 File). Following a similar process used by Lu et al., [17] reviewers identified each study's equity-oriented classification based on extracted data as one of three categories:

a) Not equity-oriented: Study does not have an explicit equity-oriented objective AND no PROGRESS-Plus factors are used in the data analysis/results as stratifying variables

b) Equity-informed: PROGRESS-Plus factors are used in the data analysis/results as stratifying variables BUT the primary outcome is not reported for a PROGRESS-Plus defined group

c) Equity-focused: The study objective explicitly pertains to equity AND the entire study population is defined by a PROGRESS-Plus factor or a PROGRESS-Plus factor is used to stratify reporting of the primary outcome

This classification system emphasizes that only reporting on PROGRESS-plus factors in a study is not sufficient for it to be equity-oriented (for example, although most studies regularly report on participant sex, most are not designed or expected to impact gender/sex equity). We defined an equity-oriented study as one that either intentionally uses

PROGRESS-Plus factors to better understand their data (equity-informed) or sets out to better understand or impact an explicitly identified health inequity (equity-focused).

To better understand the current approaches to incorporating health equity into A&F trials, we completed a sub-analysis for articles that were categorized as equity-focused. For these articles, extracted intervention data was further analyzed to determine if and how equity was considered in the A&F process of the trial intervention. Specifically, we looked at whether PROGRESS-Plus factors (different from the A&F trial's outcome or population of focus) were explicitly identified in the 'Audit' or 'Feedback' portions of the A&F trial.

Descriptive statistics were used to organize our data by equity-oriented classification and PROGRESS-Plus factors.

## Results

### Equity oriented classification of included articles

Described in detail elsewhere, the updated systematic review retrieved 20,171 articles, of which, 276 were included in the systematic review [9]. From these, 271 articles were included in this secondary analysis. The five excluded articles had an identical author, study design and intervention as an already included article, but with a different primary outcome of focus. Because of the scope of this secondary analysis, in cases where identical articles reported on multiple primary outcomes, we considered the articles for equity-orientation only once, as a whole, not for each outcome separately. A detailed list of general characteristics of included studies is presented elsewhere [9]. From the 271 articles included within this analysis, 44% of trials were classified as not equity-oriented (n = 120) and 56% as equity-oriented (n = 151). Within equity-oriented trials, 35% were classified as equity-informed (n = 95) and 21% as equity-focused (n = 56) (Fig 1). Fig 2 shows the trend of trial classifications over time, with the earliest included study from 1980 to the latest in 2020. The number of equity-focused and equity-informed trials has increased over time, with the greatest proportional increase seen in equity-focused A&F trials, with the majority (n = 32, 57%) being published after 2010.

### Equity oriented characteristics reported within included articles

As expected, given our classification criteria, PROGRESS-Plus factors are reported more frequently throughout article sections in equity-informed and equity-focused articles compared to articles that were not equity-oriented (Table 2). Only two equity-focused articles described an equity-oriented framework approach: the chronic care model [20], and the PROGRESS-Plus equity framework [29]. (Table 2)

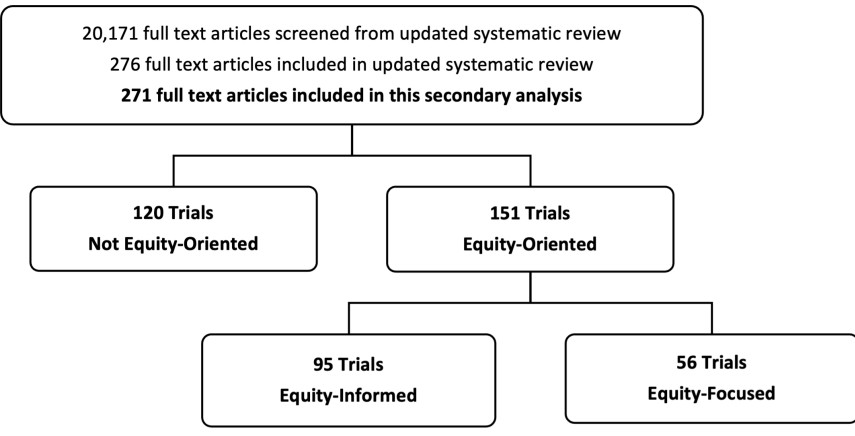

**Fig 1. PRISMA flow diagram.**

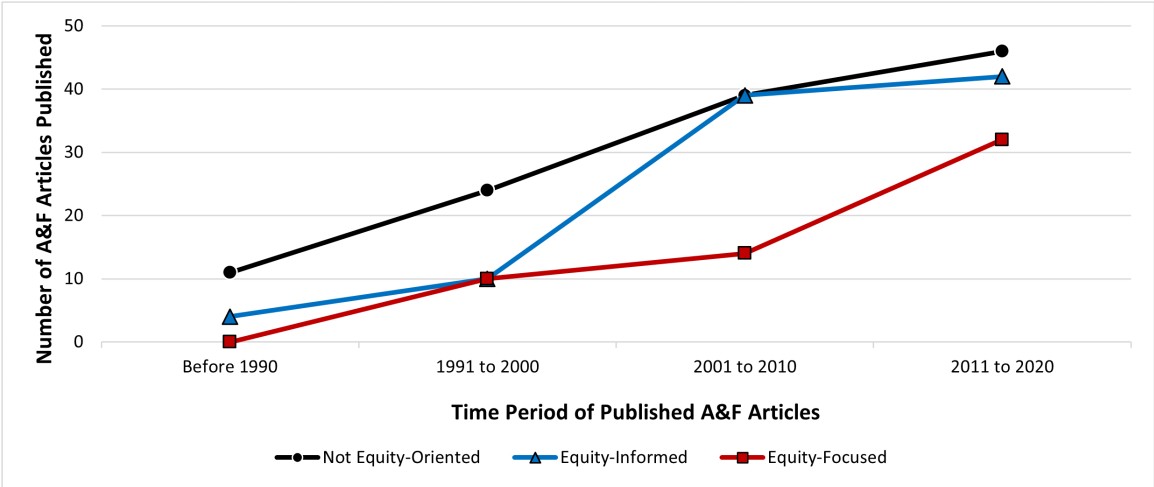

**Fig 2. Line graph depicting the trend of equity-relevant A&F trials, over time.** Horizontal axis represents year of publication categories. Vertical axis represents frequency of trials published. Legend: Black/Circle: not equity-oriented, Blue/Triangle: equity-informed, and Red/Square: equity-focused.

Frequency of specific PROGRESS-Plus factors applied in equity-informed and equity-focused article sections are illustrated in Fig 3. The Plus factors, including age and insurance, were present across both equity-informed and equity-focused articles. This was followed by place of residence (e.g., rurality) and gender/sex. Religion was reported the least (5% in equity-focused). (Fig 3)

Upon further analysis of the 56 equity-focused articles, we identified three articles that explicitly reported how equity was considered in their A&F process (Table 3). Two articles highlighted PROGRESS-Plus factors as part of patient data in their feedback or organized patient data based on factors including substance use (plus), age (plus), and gender/sex. [30,31] Based on O'Neill's definitions of the PROGRESS-Plus factors, gender/sex and personal characteristics including substance use and age may limit a person's ability to manage their health or obtain health-care and may be associated with stigmatization and discriminatory behaviours. Additionally, gender/sex may indicate differential access to healthcare services or health risks [14] One study broadly stated that their feedback included a discussion about "structural, organizational, and social barriers to change" which could be inclusive of any or all of the PROGRESS-Plus factors. [32] (Table 3)

## Discussion

We found that the number of equity-focused trials evaluating A&F has increased over the last three decades, despite representing a minority of A&F trials, overall. In our review, only two trials used an equity-oriented framework in their approach (3.6%), and only three studies explicitly reported how equity was embedded in their A&F process (5.5%).

Our approach follows the secondary analysis of diabetes quality improvement trials reported by Lu et al. in 2018. [17] However, they identified double the number of equity-focused trials compared to equity-informed, whereas we saw the opposite (Fig 1). Additionally, our study cut across multiple health issues, whereas Lu et al. focused solely on one health condition. [17] The inclusion of multiple health issues could contribute to this difference in our findings. Despite these differences, there are several similarities between our findings. We found an increase in equity-oriented studies over time (Fig 2), and that age and gender/sex were the most cited PROGRESS-Plus factors (Fig 3), which were similarly reported by Lu and colleagues. [17] Furthermore, we found that 36% of equity-focused trials used PROGRESS-Plus factors to stratify outcomes (Table 2) – higher than the 23% identified by Lu et al. [17]

Table 2. Overview of Reported Equity-Oriented Characteristics by Article Section.

| Reported Equity-Oriented Characteristics Categorized into Article Sections | Classification, N = 271 (Count, %) | | | Text Examples from Equity-Focused Articles |
|---|---|---|---|---|
| | Not Equity-Oriented n = 120 | Equity-Informed n = 95 | Equity-Focused n = 56 | |
| **Title** | | | | |
| Equity-related verbiage identified | 6 (5.0) | 9 (9.4) | 31 (55.4) | "Web-Based Just-in-Time Information and Feedback on Antibiotic Use for Village Doctors in Rural Anhui, China" [18] |
| **Abstract** | | | | |
| PROGRESS-Plus related outcome or participant demographic identified | 27 (22.7) | 41 (42.7) | 46 (82.1) | "Sixty-nine primary health care practices with 119,910 patients aged older than or equal to 65 were randomized" [19] |
| **Background** | | | | |
| Description of equity-oriented framework or approach | 0 (0.0) | 0 (0.0) | 2 (3.6) | "We hypothesized that barriers to adherence would be amenable to a health system intervention, based on elements of the chronic care model" [20] |
| Description of anticipated difference of intervention effects on subgroups as defined by PROGRESS-Plus | 14 (11.8) | 19 (19.8) | 44 (78.6) | "Based on seven separate population-based surveys, it was found that while more than 90% of non-Hispanic white women ages 50-70 had a regular source of medical care and 46-76% had a clinical breast exam within the past year, only 25-41% had a mammogram. The situation was even worse for less educated, older, and poorer women." [21] |
| **Objective** | | | | |
| Explicitly pertains to equity | 1 (0.8) | 4 (4.2) | 48 (85.7) | "To test the benefit of clinician and family directed decision support, delivered by using the [electronic health record] and telephone, on receipt of [human papillomavirus] vaccine for adolescent girls." [22] |
| **Recruitment Process*** | | | | |
| Specifies process to increase enrollment of groups characterized by PROGRESS-Plus factors | N/A* | 7 (12.5) | 11 (19.6) | "All township hospitals across the two counties were considered eligible, apart from the two situated in each county centre because their better staff capacity, equipment, and close proximity to the county general hospital made their practice quite different from that in the other township hospitals." [23] |
| **Participant Flow*** | | | | |
| Specifies differential actual enrollment related to PROGRESS-Plus factors | N/A* | 10 (10.5) | 6 (10.7) | "Comparisons of these 95 physicians to the 132 other eligible nonparticipating physicians indicate … The difference in participation then is wholly attributable to type of medical school rather than race." [21] |
| Specifies differential attrition related to PROGRESS-Plus factors | N/A* | 2 (2.1) | 1 (1.8) | "Those participants that were missing at the second audit were more likely to be male (p=< 0.001) and be older (p = 0.003) than participants available for follow-up at the second audit." [24] |
| Specifies concerns related to adherence to the intervention | N/A* | 6 (6.3) | 3 (5.4) | Differences in adherence also occurred between age groups, being best in children ≤10 years and worst in the group aged 21–40 years (P < 0.001). Adherence did not differ between male and female patients." [20] |
| **Eligibility Criteria** | | | | |
| Participant eligibility is defined by a PROGRESS-Plus factor | 41 (34.2) | 54 (56.8) | 37 (66.1) | "Eligible patients included men (age 50–75 years) and women (age 40–75 years)... spoke English; were identified as having limited [health literacy] via the Rapid Estimate of Adult Literacy in Medicine [REALM ≤60 equivalent to ≤8th grade]" [25] |
| **Results** | | | | |
| Reported participant characteristics defined by a PROGRESS-Plus factor | 86 (71.7) | 87 (91.6) | 54 (96.4) | "Of the 25 [Pediatric Research Consortium] practices, 18 primarily suburban practices not involved in resident teaching and all 4 urban, resident teaching practices participated in the study." [22] |
| PROGRESS-Plus factor used for analysis only and not stratification | 2 (1.7) | 60 (63.2) | 10 (17.9) | "Adjustment for case mix between hospitals was done by including patient-level factors in the model, namely sex and age." [26] |

*(Continued)*

**Table 2.** (Continued)

| Reported Equity-Oriented Characteristics Categorized into Article Sections | Classification, N=271 (Count, %) | | | Text Examples from Equity-Focused Articles |
|---|---|---|---|---|
| | **Not Equity-Oriented n=120** | **Equity-Informed n=95** | **Equity-Focused n=56** | |
| PROGRESS-Plus factor used to stratify outcomes | 1 (0.8) | 28 (29.4) | 20 (35.7) | "Results were consistent across predefined subgroups based on sex, age, clinic size... pregnancy, and tuberculosis at eligibility." [27] |
| **Discussion*** | | | | |
| Applicability, generaliz-ability, or external validity are discussed in reference PROGRESS-Plus factors | N/A* | 24 (25.3) | 26 (46.4) | "While our study was generalizable in terms of geography and community size, we had an over-representation of chain affiliated and for-profit long-term care homes compared with provincial averages. Not-for-profit homes have been associated with higher quality of care, although multifacility chains may have greater resources to facilitate implementation of clinical practice guidelines." [28] |

*Extracted only for equity-informed and equity-focused articles.

| PROGRESS-Plus Factor | Article Sections (count, %) | | | | | | | |
|---|---|---|---|---|---|---|---|---|
| | **Objectives** | | **Reported Participant Characteristics** | | **Used in Stratification of Outcomes** | | **Defined Entire Data Set** | |
| | *EI* | *EF* | *EI* | *EF* | *EI* | *EF* | *EI* | *EF* |
| **P**lace of residence | 0 | 20 (35.7) | 26 (27.4) | 12 (21.4) | 5 (5.3) | 2 (3.6) | 1 (1.1) | 14 (25.0) |
| **R**ace/ Ethnicity/ Culture/ Language | 0 | 1 (1.8) | 25 (26.3) | 15 (26.8) | 10 (10.5) | 5 (8.9) | 0 | 1 (1.8) |
| **O**ccupation | 0 | 16 (28.6) | 10 (10.5) | 8 (14.3) | 1 (1.1) | 0 | 0 | 0 |
| **G**ender/sex | 0 | 7 (12.5) | 72 (75.8) | 42 (75.0) | 12 (12.6) | 11 (19.6) | 7 (7.4) | 4 (7.1) |
| **R**eligion | 0 | 0 | 0 | 3 (5.4) | 0 | 0 | 0 | 0 |
| **E**ducation | 0 | 1 (1.8) | 6 (6.3) | 10 (17.9) | 0 | 3 (5.4) | 0 | 0 |
| **S**ocioeconomic status | 0 | 2 (3.6) | 5 (5.3) | 6 (10.7) | 0 | 0 | 0 | 1 (1.8) |
| **S**ocial Capital | 0 | 0 | 4 (4.2) | 4 (7.1) | 1 (1.1) | 1 (1.8) | 0 | 0 |
| **P**lus* | 5 (5.3) | 29 (51.8) | 77 (81.1) | 49 (87.5) | 22 (23.2) | 16 (28.6) | 3 (3.2) | 2 (3.6) |
| Age | 3 (3.2) | 24 (42.9) | 72 (75.8) | 47 (83.9) | 20 (21.1) | 15 (26.8) | 3 (3.2) | 2 (3.6) |
| Insurance | 1 (1.1) | 3 (7.1) | 15 (15.8) | 14 (25.0) | 5 (5.3) | 2 (3.6) | 1 (1.1) | 1 (1.8) |

0-20% 20.1-40% 40.1-60% 60.1-80% 80.1-100%

*EI: equity-informed, EF: equity-focused*

**Fig 3. Heatmap of Frequency of PROGRESS-Plus factors applied in equity-informed (n = 95) and equity-focused articles (n = 56).** *Plus categories include factors other than age and insurance including disability, co-morbidities, and mental health.

**Table 3. Sub-Analysis of Equity Embedded in Feedback within Equity-Focused Articles.**

| Article | Population, Intervention & Comparison Groups | Audit & Feedback *Highlighting Reported PROGRESS Plus Factor & Author Rationale*\* |
|---|---|---|
| Soumerai 1998 [30] | Population: Appropriate therapies for Acute Myocardial Infarction Intervention: 1-day meeting with opinion leaders to promote practice change with feedback provided on performance Opinion leaders' local tools and resources to influence change: presentations, administrative support, education brochures Comparison: Usual care with mailed feedback on performance only | Feedback included performance of hospital on guideline adherence including **patient baseline use of study drugs (by age and sex; e.g., proportion of eligible elderly patients receiving aspirin)** for each trial hospital and utilization rates of non-study drugs. *[Age, Gender/Sex: authors reported that previous studies identified age and gender/sex as variables that predicted use of study drugs]* |
| Mitchell 2005 [31] | Population: Older patients (65–79 years) with hypertension Intervention: 1) Audit only included 'rule of halves feedback' 2) Audit plus risk included 'rule of halves feedback' and risk feedback Comparison: No feedback | Rules of halves: Practices received anonymized patient data including reports of blood pressure, normal or high blood pressure receiving antihypertensive treatment, and **additional risk factors: smoking,** diabetes, previous stroke. Risk: patient-specific list ranked according to absolute risk of death from stroke in the next 10 years. **Patients without smoking status were given two scores, one as someone who smokes, and one as someone who does not.** *[Plus: Substance use: authors did not provide a rationale]* |
| Pettersson 2011 [32] | Population: Infections or inappropriate use of antibiotics for nursing home residents Intervention: Educational group presentation and discussions with prescribers working in nursing homes Comparison: No intervention | Prescribers were presented with feedback, references to available guidelines, and **discussions about structural, organizational, and social barriers to change.** *[All: authors did not provide a rationale]* |

A&F: Audit and Feedback. \*Equity was highlighted only if the emphasis was different from the original article objective or patient population (e.g., if the population was elderly patients, then feedback specific to elderly patients was not included; however, if the trial population was not specific to elderly patients and trial developers decided to provide feedback about elderly patients as an additional value, this feedback was included).

We acknowledge some subjectivity in how we identified PROGRESS-Plus factors. For example, we debated whether nursing home populations should be considered within the age (plus) factor and ultimately decided to include them if the population was described as being at a particular disadvantage (e.g., an institutionalized care setting). Relatedly, the process of categorizing trials as 'not equity-oriented', 'equity-informed', or 'equity-focused' was complex. It was sometimes difficult to identify equity-orientation as a goal of the intervention versus as a feature of or confounded with the clinical topic of interest. We developed a set of criteria, as described in our methods, for selecting a category, yet there were times when each reviewer chose different categories, and the final decision required team discussion. For example, an article may have used a PROGRESS-Plus factor to stratify reported outcomes which, early in the process, would have led us to classify it as equity-focused. However, in some circumstances, there was a lack of intentionality or context for the

stratification of outcomes, and this led to a classification of equity-informed instead. These differences and discussions led to some refinement of the criteria over time. Although we reviewed all classifications at the end of the process, it is possible that the evolution of the criteria and our thinking over time resulted in some discrepancies in classification. Furthermore, we did not assess differences in efficacy between equity-oriented and not-equity oriented trials. However, such an analysis would be extremely complex and at high risk of bias due to the variability within equity-oriented studies regarding intervention components, equity concerns, and populations.

## Potential impact & future directions

Our findings suggest an opportunity for A&F trial developers to improve how they embed and report on equity. Trials that aim to be more inclusive can follow trends within the equity-focused trials found in this study, including explicitly identifying equity-related factors within study objectives, considering how factors like the social determinants of health may provide a health-related (dis)advantage and affect study outcomes within study backgrounds, and conducting sub-group analyses for distinct equity-related factors that may be relevant to the population of interest. We identified only four trials that explicitly reported feedback that was inclusive of equity considerations extending beyond the trials' population of focus. There is a significant opportunity for A&F intervention designers to create formal pathways to embed equity within feedback, to facilitate better health outcomes for disadvantaged populations. This secondary analysis was a first step in exploring if A&F interventions were considering equity and how; we hope this work encourages others to do additional analyses to further outline differences in efficacy between trials that are equity-oriented and those that are not.

## Conclusion

Despite the substantial evidence base in support of A&F to improve care in general, there remains limited evidence on whether (or how) to use A&F to address health inequities. This study reviewed A&F trials to examine the extent to which trials consider health equity over time and provides examples of ways to embed equity in implementation trial designs, A&F processes, and reporting that may be replicated in future trials.

## Supporting information

**S1 Table. PRISMA-Equity Checklist.**
(DOCX)

**S2 Table. Definition of PROGRESS-Plus Factors.** Adapted from Lu et al. (2018) Supplemental info.
(DOCX)

**S1 File. Data Extraction Form.**
(DOCX)

## Acknowledgments

We thank all colleagues who contributed to the systematic review. Thank you to Jesmin Antony and Sharlini Yogasingam for their contributions to this work. This work is a secondary analysis of the updated Cochrane A&F review by Ivers et al. [9] Noah Ivers is supported by a Canada Research Chair in Implementation of Evidence-based Practice and by the Department of Family and Community Medicine at Women's College Hospital and the University of Toronto.

## Author contributions

**Conceptualization:** Camille Williams, Ambreen Sayani, Lynne Moore, Aisha Lofters, Noah M Ivers.
**Data curation:** Camille Williams, Noah M Ivers.

**Formal analysis:** Zeenat Ladak, Camille Williams, Tolulope Ojo, Camille Renee, Aranee Senthilmurugan, Thomas A Willis, Victor C Rentes, Armaghan Dabbagh, Heather A Shepherd, Tasneem Khan, Janyce Gnanvi, Mary Carter.

**Methodology:** Zeenat Ladak, Camille Williams, Noah M Ivers.

**Project administration:** Camille Williams.

**Supervision:** Noah M Ivers.

**Visualization:** Zeenat Ladak.

**Writing – original draft:** Zeenat Ladak, Camille Williams.

**Writing – review & editing:** Tolulope Ojo, Camille Renee, Aranee Senthilmurugan, Thomas A Willis, Victor C Rentes, Armaghan Dabbagh, Heather A Shepherd, Tasneem Khan, Janyce Gnanvi, Mary Carter, Ambreen Sayani, Lynne Moore, Aisha Lofters, Noah M Ivers.

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
