## [Decision Letter · Decision Letter 0]

20 Oct 2025

Dear Dr. Ladak,

We look forward to receiving your revised manuscript.

Kind regards,

Kiyan Heybati, MD, MSc

Academic Editor

PLOS ONE

**Journal Requirements:**

1. When submitting your revision, we need you to address these additional requirements. Please ensure that your manuscript meets PLOS ONE's style requirements, including those for file naming. The PLOS ONE style templates can be found at https://journals.plos.org/plosone/s/file?id=wjVg/PLOSOne_formatting_sample_main_body.pdf and https://journals.plos.org/plosone/s/file?id=ba62/PLOSOne_formatting_sample_title_authors_affiliations.pdf 2. Please include captions for your Supporting Information files at the end of your manuscript, and update any in-text citations to match accordingly. Please see our Supporting Information guidelines for more information: http://journals.plos.org/plosone/s/supporting-information. 3. If the reviewer comments include a recommendation to cite specific previously published works, please review and evaluate these publications to determine whether they are relevant and should be cited. There is no requirement to cite these works unless the editor has indicated otherwise. 

Reviewers' comments:

**Comments to the Author**

1. Is the manuscript technically sound, and do the data support the conclusions?

Reviewer #1: Yes

Reviewer #2: Yes

2. Has the statistical analysis been performed appropriately and rigorously?

Reviewer #1: Yes

Reviewer #2: Yes

3. Have the authors made all data underlying the findings in their manuscript fully available?

Reviewer #1: Yes

Reviewer #2: Yes

4. Is the manuscript presented in an intelligible fashion and written in standard English?

Reviewer #1: Yes

Reviewer #2: Yes

**Reviewer #1:** I have carefully read through the manuscript and applaud the authors for the work put into this. The topic is essential in recent times, as it addresses equity through quality of care, and its importance to the health systems cannot be overemphasised enough.

Please consider the following revisions.

In the methods section under the study selection criteria, all dates put together are confusing, and I cannot be sure when the review started and ended. Also, the period when the articles were retrieved or the period when the articles included were considered is not clear.

Again, under the results, you reported 276 articles included in the main review, but you did not give the total number of articles that were retrieved.

**Reviewer #2:** General comments:

The main finding of the study is the description of audit and feedback (A&F) trials that include components of equity in the feedback content. It would have been helpful if the study had provided comparative information on how these trials differ from A&F trials without equity components.

It would also be helpful to offer a practical definition of what constitutes an A&F initiative with equity components. For example, age distribution may reflect a potential equity issue, or it may simply represent the inherent characteristics of the underlying disease studied or the eligible patient population.

The study does not provide information whether the equity-focused or informed A&F interventions lead to changes in provider behavior or components of behavior. An approach could be to frame this as a meta-epidemiological analysis, or a qualitative synthesis, comparing A&F trials with and without equity components in terms of outcomes such as provider behavior or even patient-level health outcomes.

Minor comments:

Introduction (Lines 79–81):

It would be helpful to specify how equity factors contribute to systematic health inequalities. For example, is it through differences in availability of services, accessibility, exposure to different risk factors, or other mechanisms?

Methods:

In Table 1, based on the stated objective, the appropriate comparator should be A&F trials without equity components.

Discussion:

It would be helpful to clearly state whether it is assumed that A&F interventions with equity components demonstrate different efficacy in addressing equitable outcomes compared to standard A&F. Or does the efficacy of A&F itself depend on equity factors (i.e., an interaction effect)?

**Do you want your identity to be public for this peer review?** For information about this choice, including consent withdrawal, please see our Privacy Policy

Reviewer #1: No

Reviewer #2: No

---

## [Author Response · Author response to Decision Letter 1]

12 Nov 2025

Please see attached Response to Reviewers document. Thank you.

---

## [Decision Letter · Decision Letter 1]

7 Dec 2025

Exploring equity in audit and feedback trials: Secondary analysis of a systematic review

PONE-D-25-32703R1

Dear Dr. Ladak,

We’re pleased to inform you that your manuscript has been judged scientifically suitable for publication and will be formally accepted for publication once it meets all outstanding technical requirements.

Kind regards,

Kiyan Heybati

Academic Editor

PLOS One

Additional Editor Comments (optional):

Reviewers' comments:

Reviewer's Responses to Questions

**Comments to the Author**

Reviewer #1: All comments have been addressed

Reviewer #2: All comments have been addressed

2. Is the manuscript technically sound, and do the data support the conclusions?

Reviewer #1: Yes

Reviewer #2: Yes

3. Has the statistical analysis been performed appropriately and rigorously?

Reviewer #1: Yes

Reviewer #2: Yes

4. Have the authors made all data underlying the findings in their manuscript fully available?

Reviewer #1: Yes

Reviewer #2: Yes

5. Is the manuscript presented in an intelligible fashion and written in standard English?

Reviewer #1: (No Response)

Reviewer #2: Yes

Reviewer #1: (No Response)

Reviewer #2: Thank you for the responses. I have no further comments.

One point for consideration:

As categorizing studies (as equity-oriented or not) was largely a subjective consensus process, and only a small number of studies applied an equity framework, these findings imply a need for standard reporting of equity considerations in such interventions. This can help highlight existing gaps.

**Do you want your identity to be public for this peer review?** For information about this choice, including consent withdrawal, please see our Privacy Policy

Reviewer #1: No

Reviewer #2: No

---

## [Editor Report · Acceptance letter]

PONE-D-25-32703R1

PLOS One

Dear Dr. Ladak,

I'm pleased to inform you that your manuscript has been deemed suitable for publication in PLOS One. Congratulations! Your manuscript is now being handed over to our production team.

Kind regards,

on behalf of

Dr. Kiyan Heybati

Academic Editor

PLOS One